# Metabolic View on Human Healthspan: A Lipidome-Wide Association Study

**DOI:** 10.3390/metabo11050287

**Published:** 2021-04-30

**Authors:** Justin Carrard, Hector Gallart-Ayala, Denis Infanger, Tony Teav, Jonathan Wagner, Raphael Knaier, Flora Colledge, Lukas Streese, Karsten Königstein, Timo Hinrichs, Henner Hanssen, Julijana Ivanisevic, Arno Schmidt-Trucksäss

**Affiliations:** 1Division of Sports and Exercise Medicine, Department of Sport, Exercise and Health, University of Basel, Birsstrasse 320B, CH-4052 Basel, Switzerland; justin.carrard@unibas.ch (J.C.); denis.infanger@unibas.ch (D.I.); jonathan.wagner@unibas.ch (J.W.); raphael.knaier@unibas.ch (R.K.); lukas.streese@unibas.ch (L.S.); k.koenigstein@unibas.ch (K.K.); timo.hinrichs@unibas.ch (T.H.); henner.hanssen@unibas.ch (H.H.); 2Metabolomics Platform, Faculty of Biology and Medicine, University of Lausanne, Quartier UNIL-CHUV, Rue du Bugnon 19, CH-1005 Lausanne, Switzerland; hector.gallartayala@unil.ch (H.G.-A.); tony.teav@unil.ch (T.T.); 3Division of Sports Science, Department of Sport, Exercise and Health, University of Basel, Birsstrasse 320B, CH-4052 Basel, Switzerland; flora.colledge@unibas.ch

**Keywords:** healthspan, healthy population study, metabolic phenotyping, lipidomics, serum lipid signature

## Abstract

As ageing is a major risk factor for the development of non-communicable diseases, extending healthspan has become a medical and societal necessity. Precise lipid phenotyping that captures metabolic individuality could support healthspan extension strategies. This study applied ‘omic-scale lipid profiling to characterise sex-specific age-related differences in the serum lipidome composition of healthy humans. A subset of the COmPLETE-Health study, composed of 73 young (25.2 ± 2.6 years, 43% female) and 77 aged (73.5 ± 2.3 years, 48% female) clinically healthy individuals, was investigated, using an untargeted liquid chromatography high-resolution mass spectrometry approach. Compared to their younger counterparts, aged females and males exhibited significant higher levels in 138 and 107 lipid species representing 15 and 13 distinct subclasses, respectively. Percentage of difference ranged from 5.8% to 61.7% (females) and from 5.3% to 46.0% (males), with sphingolipid and glycerophophospholipid species displaying the greatest amplitudes. Remarkably, specific sphingolipid and glycerophospholipid species, previously described as cardiometabolically favourable, were found elevated in aged individuals. Furthermore, specific ether-glycerophospholipid and lyso-glycerophosphocholine species displayed higher levels in aged females only, revealing a more favourable lipidome evolution in females. Altogether, age determined the circulating lipidome composition, while lipid species analysis revealed additional findings that were not observed at the subclass level.

## 1. Introduction

The ageing of the world’s population contributes largely to the growing prevalence of non-communicable diseases [1]. Simultaneously, disease-specific healthcare approaches tend to foster survival with chronic diseases and disabilities rather than contributing to a longer healthy life, referred to as healthspan [1,2,3,4,5,6]. As a result, ageing and related morbidities impose an increasing socio-economic burden on nations [7]. Thus, the World Health Organization (WHO) launched the concept of healthy ageing, which focuses on the preservation of physiological functions across the course of life to increase healthspan [1,2]. To implement this concept into clinical practice, new phenotyping tools that could capture metabolic individuality and stratify patients with respect to potential health decline and disease onset, are needed [8,9].

Total cholesterol, low-density lipoprotein cholesterol (LDL-C), high-density lipoprotein cholesterol (HDL-C) and triglycerides are still the main lipid markers used to assess cardiometabolic risk in clinical medicine [10,11]. However, the human plasma lipidome is estimated to consist of hundreds of thousands of lipid species, which are extremely diverse in both chemical structures and biological functions [12]. Following the technological advances in mass spectrometry and bioinformatics, lipidomics has evolved into a high-throughput approach to allow for an in-depth investigation of lipid metabolism—at the species level [13,14]. In light of their involvement in numerous biological processes, lipids are believed to act as key modulators of health, ageing and pathogenesis of cardiometabolic diseases [12,15,16]. Indeed, lipids are essential to energy storage, cell structure and molecular signalling [12,17]. They are the main constituents of plasma membranes, where they modulate receptor activity and vesicular trafficking [18,19]. The term bioactive lipid is used to designate lipid species in which a change in abundance leads to functional repercussions [20]. Sphingolipids in general, and ceramides in particular, form an important category of bioactive lipids, as they modulate numerous critical biological processes and have been implicated in many cardiometabolic, oncological and neurodegenerative disorders [20,21,22]. For instance, ceramides located at the surface of low-density lipoproteins (LDL) drive their aggregation and transcytosis through the endothelium as well as uptake into macrophages, which leads to foam cell formation and vascular inflammation [23,24,25]. Remarkably, plasma ceramides are strong predictors of cardiovascular death in patients with and without coronary artery disease [26,27,28,29]. Therefore, ceramides could become the “new cholesterol” in daily clinical practice [30].

In view of these findings, ‘omic-scale lipid analysis shows potential for improving clinical patients’ stratification [13,14,31]. However, prior to profiling patients, clinical studies investigating healthy populations are necessary to decipher the relations between circulatory lipid species and key biological determinants, such as age and sex. This is essential to further elucidate the role of the many lipid species in health maintenance. Investigating healthy aged individuals enables the disentanglement of age- from disease-related metabolic changes, which is rarely done as most human ageing studies enrolled patients already suffering from chronic diseases or centenarians of unspecified or poorly characterised health status [32,33,34,35,36,37,38,39,40,41,42,43,44,45,46]. Moreover, the few studies which claimed to have examined the metabolic profile of healthy individuals did not report on or adjust for physical activity levels [47,48,49,50]. Considering that physical inactivity has been recognised as the fourth leading cause of death worldwide, it is coherent to consider physical activity as a key determinant of human health [51,52].

This cross-sectional population-based study had two aims. Firstly, to acquire the serum lipid profile of clinically healthy humans in their twenties and seventies using an untargeted liquid chromatography high-resolution mass spectrometry approach. Secondly, to identify lipid species associated with age and sex [53]. To this end, sera of a subset of the COmPLETE-Health study, composed of young (20–29 years) and aged (70–79 years) well characterised, clinically healthy females and males, were investigated [54].

## 2. Results

### 2.1. Characteristics of the Clinically Healthy Participants

The examined subset consisted of 73 young (25.2 ± 2.6 years, 44% female) and 77 aged (73.5 ± 2.3 years, 48% female) participants of the COmPLETE-Health study. Participants underwent objective physical activity tracking, blood sampling and lipid phenotyping (Figure 1A). Categorised by age and sex, each group displayed, on average, normal to high-normal blood pressure, and normal body mass index, triglycerides and glycated haemoglobin (HbA1c) levels (Table 1) [10,55,56]. The aged participants exhibited slightly elevated mean LDL-C levels compared to the maximal recommended value of 3 mmol/L for low-risk patients [10]. In each group, the mean fasting time prior to blood sampling was 5.6 h at least. All participants fulfilled the WHO recommendations in terms of daily physical activity time (moderate-to-vigorous physical activity ranging from 39.1 to 321.4 min/day, with a mean of 158.9 ± 57.1 min/day) [52]. Clinical data distribution is presented in Appendix A.

### 2.2. Lipid Signature of Clinically Healthy Young and Aged Phenotypes

As shown in Table 2, 218 lipid species belonging to 16 distinct lipid subclasses were identified with a high level of confidence (using accurate *m/z* ratio and MS/MS fragmentation pattern). Principal Component Analysis (PCA) revealed that young and aged participants displayed two distinct serum lipid signatures (PC1 42.48%, PC2 12.85%, Figure 1B and Appendix A), while differential analysis showed that 179 lipid species exhibited higher levels in aged compared to young participants (Figure 1C and Appendix A). After adjustment for body fat (%), statins intake, HbA1c (%), daily total physical activity (min), blood sampling time and fasting time prior to blood sampling, 12 of 16 lipid subclasses were significantly and positively associated with age (Figure 2 and Appendix A). The strongest associations were observed for lyso-alkyl-glycerophosphoethanolamines (LPE-O, β coefficient = 1.49, Benjamini–Hochberg (BH) *p*-value ≤ 0.0001), glycerophosphoinositols (PI, β coefficient = 1.43, BH *p*-value ≤ 0.0001), ceramides (Cer, β coefficient = 1.41, BH *p*-value ≤ 0.0001) and sphingomyelins (SM, β coefficient = 1.40, BH *p*-value ≤ 0.0001). On the molecular level, 121 species (including 28 glycerophosphocholines, PC, and 23 SMs) were significantly and positively associated with age (Appendix A). The strongest associations were observed for PC(16:0_18:0) (β coefficient = 1.67, BH *p*-value ≤ 0.0001) and SM(37:2;3) (β coefficient = 1.67, BH *p*-value ≤ 0.0001). Strikingly, no lipid subclass or species was significantly and negatively associated with age. Daily total physical activity did not display any significant association with any lipid species or subclass (Appendix A).

Post-hoc tests were calculated to determine and compare the estimated marginal means of each lipid in aged and young participants within both sexes, using the emmeans R-package (version 1.4.8) [57]. They revealed that the subclasses SM, Cer, PI, glycosphingolipids (GSL), PC and lyso-glycerophosphoinositols (LPI) displayed, in both sexes, significantly higher levels in aged compared to young participants (Appendix A). In females, 138 lipid species (including 32 PCs and 23 SMs) showed significantly higher levels in aged compared to young participants, with the percentage of difference ranging from 5.8% to 61.7% (Figure 3A and Appendix A). In males, 107 lipid species (including 28 PCs and 23 SMs) exhibited significantly higher levels in aged compared to young subjects, with the percentage of difference ranging from 5.3% to 46.0% (Figure 3B and Appendix A). In both females and males, SM(37:2;3) displayed the greatest percentage of difference (females: 61.7%, BH *p*-value ≤ 0.0001; males: 46.0%, BH *p*-value ≤ 0.0001). No lipid subclass or species was significantly decreased with age in both females and males.

Regarding the saturation level, age was negatively associated with the monounsaturated fatty acid (MUFA) to polyunsaturated fatty acids (PUFA) ratio for PI (Appendix A). Post-hoc tests revealed that the MUFA/PUFA ratios for PI and lyso-glycerophosphoethanolamines (LPE) were significantly lower in aged compared to young females, while the MUFA/PUFA ratio for Cer was significantly higher in aged compared to young males (Appendix A). Concerning glycerophosphocholines (PC) to glycerophosphoethanolamines (PE) ratios, age was negatively associated with the ratio alkyl-glycerophosphocholines (PC-O) to alkyl-glycerophosphoethanolamines (PE-O, Appendix A). Post-hoc tests revealed that the PC-O/PE-O ratio was significantly lower in aged compared to young females (Appendix A).

Lipid ontology (LION) enrichment analysis was conducted using LION/web (version 2020.07.14), which is an online tool performing network analysis within the lipidomic dataset [58]. Containing a library of >50,000 lipid species, LION/web goes beyond classical analysis solely based on lipid nomenclature by providing insights whether specific physicochemical properties, biological functions or cellular localisation are enriched in a given group or condition of interest. Thus, LION/web aims to facilitate the interpretation of complex lipidomic datasets within a biological context. All 218 lipid species could be matched with a LION ID within the LION library. Twenty-three LION terms were significantly enriched in aged compared to young females (Figure 4A and Appendix A), with “1-alkyl,2-acylglycerophosphoethanolamines” (PE-O) being the most enriched LION term (15 matches, *q*-value ≤ 0.0001). In males, nine LION terms were significantly enriched in the aged compared to the young (Figure 4B and Appendix A), with “sphingolipids” being the most enriched LION term (35 matches, *q*-value ≤ 0.0001).

### 2.3. Age-Related Sex Differences in the Circulatory Lipidome Composition

The subclasses diglycerides (DG), TG, alkenyl-glycerophosphoethanolamines (PE-P) and LPE-O as well as 23 species (including 17 TGs) were significantly and positively associated with the male sex (Figure 2, Appendix A). Conversely, SM(32:2;2), SM(38:2;2), PC(16:0_16:1), PE(16:0_22:6), PI(34:1) and PC(O-34:2) were significantly and negatively associated with the male sex (Appendix A).

Within the young participants, 15 lipid species (including six SMs and three PCs) exhibited higher levels in females compared to males (Appendix A). Conversely, the subclasses DG, TG, PE-P, LPE-O and lyso-glycerophosphocholines (LPC), as well as 34 species (including 22 TGs) displayed significantly lower levels in females compared to males (Appendix A). Within the aged participants, the subclasses PC-O and SM, as well as 26 lipid species (including nine SMs and seven PC-Os), displayed higher levels in females compared to males, while the subclasses TG and DG, as well as 11 species (including ten TGs), showed lower levels in females compared to males (Appendix A).

As to the saturation level, a significant and negative association was observed between the MUFA/PUFA ratio for PE-O and the male sex (Appendix A). Post-hoc tests revealed, that this ratio was significantly lower in females compared to males, in young participants only (Appendix A). The PC-O/PE-O ratio was significantly and negatively associated with the male sex (Appendix A). Again, post-hoc tests showed, that this ratio was significantly lower in females compared to males in young participants only (Appendix A).

Concerning the lipid ontology enrichment analysis, 30 LION terms were significantly enriched in young females compared to young males (Appendix A), with “membrane component” being the most enriched LION term (149 matches, *q*-value ≤ 0.0001). In the aged participants, 30 LION terms were significantly enriched in females compared to males (Appendix A) with “membrane component” also being the most enriched LION term (149 matches, *q*-value ≤ 0.0001).

## 3. Discussion

### 3.1. Unravelling Age- and Sex-Associated Lipid Signature

As all participants of the present study were free of exercise-limiting chronic diseases and fulfilled the WHO recommendations for daily physical activity, the investigated lipidome reflects clinically healthy phenotype [52]. In this way, this work could identify age- and sex-associated differences in the serum lipid profile, avoiding the confounding effects of symptomatic cardiometabolic diseases on lipid metabolism. Thereby, this study confirmed that age strongly determines the composition of the serum lipidome (Figure 2) [35,50]. Strikingly, aged females exhibited higher levels in 138 out of 218 lipid species, representing 15 distinct lipid subclasses, while aged males displayed higher levels in 107 out of 218 species, representing 13 subclasses, compared to their respective younger counterparts (Figure 3). The investigated sex-balanced sample allowed for the identification of these nuances, circumventing the selection bias towards females traditionally found in ageing studies [32,40,43].

### 3.2. Cardiometabolic Significance of the Identified Lipid Signature

As illustrated in the summarising Figure 5, aged participants were characterised by both cardiometabolically favourable and deleterious lipid features. The favourable features might indicate that aged participants benefited from a protective genetic background and/or a lifestyle to reach an advanced age without any excluding medical conditions. Indeed, as age is a major risk factor for most exclusion criteria of the present study, the selection pressure was likely greater on aged compared to young participants. As to the deleterious lipid features, they might be a sign of a beginning age-related metabolic imbalance in otherwise clinically healthy aged humans.

#### 3.2.1. Sphingolipids

Sphingolipids were highly and positively associated with age, as previously reported [33,59]. SM, which interact with cholesterol to regulate membrane trafficking and intracellular signalling, account for most sphingolipids identified in the present work [60]. As SM become biologically active once transformed into Cer by the enzyme sphingomyelin synthase, it is reasonable to think in terms of Cer when reflecting on SM biological functions [20,21,61,62]. In the present study, SM(18:1;2/24:0), its biologically active analogue Cer(18:1;2/24:0) as well as Cer(18:2;2/24:0), which have been previously reported to be cardiometabolically favourable, were found at higher levels in both aged females and males [26,29,63]. Cer(18:2;2/24:0), a sphingolipid species containing a 18:2;2 sphingadiene backbone, has been reported to be negatively associated with insulin resistance, while Cer(18:1;2/24:0) is part of the ceramide-phospholipid score for the prediction of cardiovascular risk [64,65]. Concurrently, the cardiometabolically harmful Cer(18:1;2/16:0) and Cer(18:1;2/24:1) also exhibited higher levels in aged individuals of both sexes [26,29,66,67]. The fact that structurally closely related ceramide species are associated to distinct metabolic consequences highlights the need for detailed lipid analysis at the molecular species level [26,27,28].

#### 3.2.2. Glycerophospholipids

PC species accounted for the majority of glycerophospholipids displaying significantly higher levels in aged participants of both sexes. PCs, located mainly in the outer leaflet of plasma membranes, are intrinsically linked to sphingolipids, as they are required in the *de novo* synthesis of SM from Cer [68,69]. Clinically, the favourable PC(16:0/22:5) and deleterious PC(16:0/16:0) are part of the ceramide-phospholipid score for the prediction of cardiovascular risk, while PC(36:6) has also been reported to be cardiometabolically favourable [29,63,64]. In the present work, these three species displayed higher levels within both aged females and males.

LPC, which can be generated through the breakdown of PC by phospholipases A2 and glycoprotein lecithin cholesterol acyltransferases, are considered to be the bioactive forms of PC [70]. LPC are believed to be metabolically favourable as they were shown to slowdown cholesterol synthesis and atherogenesis in macrophages [71]. In the present study, 14 LPC species exhibited higher levels with age in females, while only six LPC species showed higher levels with age in males. Specifically, LPC(16:0), which was found to be inversely associated with the incidence of cardiovascular diseases and intima-media thickness, displayed higher levels in both aged females and males [72]. Conversely, LPC(18:2), which has been negatively associated with type 2 diabetes and impaired glucose tolerance, showed higher levels with age in females only [73]. The sex-specific age-related accumulation of LPC species could be a sign of a more favourable lipid signature in aged females.

Conversely to PC, PE are mainly located in the outer leaflet of plasma membrane and in the mitochondrial inner membrane [74]. The ratio PC/PE in plasma and mitochondrial membrane composition has been shown to modulate membrane functions and mitochondrial energy production [74]. Clinically, lower PE abundance in mitochondrial membranes was observed in both Alzheimer’s and Parkinson’s diseases, while both increased and decreased hepatic PC/PE ratios were associated with non-alcoholic fatty liver disease [75]. In light of these findings, the absence of a significant difference in terms of PC/PE ratio observed between aged and young participants could be a sign of metabolic health. The biological significance of the specific PC-O/PE-O ratio, which was lower in aged compared to young females in the present work, has not yet been explained to the best of the authors’ knowledge [74].

PI have long been known as key regulators of cell physiology, yet the importance of their aliphatic chain composition has been only recently recognised [76,77,78]. PI(18:0/20:4) is the predominant species in healthy mammalian cells, while cancer cells are enriched in PI, totalling 34 and 36 carbon atoms in their aliphatic chains [79,80]. In the present work, aged participants of both sexes displayed higher levels of the presumed healthy PI(18:0/20:4) and unhealthy PI(34:1), PI(36:2) and PI(36:4). In addition, aged males showed higher levels of the presumed unhealthy PI(34:2). Again, this might be a sign for a more favourable age-related evolution of the circulating lipidome composition in females.

PC-O, PE-O and LPE-O belong to ether-glycerophospholipids [81]. In the present work, 15 PE-O, 15 PC-O and two LPE-O displayed higher levels in aged females, while only seven PE-O and six PC-O were more abundant with age in males. These results are concordant with the ones of the lipid ontology analysis, showing that the LION term “1-alkyl,2-acylglycerophosphoethanolamines” (PE-O) was enriched with age in females only (Figure 4). These sex-specific age-related differences in ether-glycerophospholipid species could once more indicate a favourable age-related evolution of the lipidome composition in females. Indeed, ethers-glycerophospholipids act, amongst others, as cellular antioxidants [81,82]. This function is closely related to peroxisomes, where they are synthesised [81]. Two lethal diseases in childhood caused by inherited deficiencies in ether-glycerophospholipid, rhizomelic chondrodysplasia punctate and Zellweger spectrum disorders, highlight the importance of ether-glycerophospholipids to human health [81]. Lower circulating levels of ether-glycerophospholipids have also been observed in patients with non-alcoholic steatohepatitis and children with type 1 diabetes [83,84]. The fact that PE-O, LPE-O and LPE are synthesised in the inner membrane of mitochondria may explain why the LION term “mitochondrion” was enriched within aged females only in the lipid ontology analysis [85]. Lastly, the terms “plasma membrane” and “membrane composition” were enriched with age in both females and males, which implies that membrane composition changes with age.

#### 3.2.3. Cholesterol Esters, Glycerolipids and Saturation Levels

The cardiometabolically favourable CE(22:6) and CE(20:5) showed higher levels in both aged females and males. Circulating CE levels have been shown to be negatively associated with cardiovascular diseases [86,87]. The acyl chains 22:6 and 20:5 likely correspond to docosahexaenoate and eicosapentaenoate, circulating levels of which are negatively associated with fatal coronary heart diseases [88,89]. Additionally, serum level of CE(22:6) has been reported to be significantly lower in patients with Alzheimer’s disease compared to cognitively healthy subjects, with lower levels of CE(22:6) corresponding to more severe dementia [90].

Ten and seven TG species displayed higher levels in aged females and males, respectively. In addition, most TG species exhibited lower levels in females compared to males (Appendix A), yet the number decreased after menopause as previously described [59,91]. While both the European Society of Cardiology and the American Heart Association consider TG as important biomarkers of cardiovascular diseases due to their association with circulating apolipoproteins B, it is not clear yet if TG are directly atherogenic [10,92]. Nevertheless, the elevated TGs levels likely indicate a beginning age-related impairment in lipid metabolism in otherwise healthy aged humans.

Finally, the absence of significant differences in MUFA/PUFA ratios between young and aged individuals in all subclasses but PI, LPE and Cer could be interpreted as a persistence of the protection against oxidative stress. Indeed, a low degree of fatty acid unsaturation in membranes and plasma has been linked with longevity [39,93,94,95]. The reason for this is that fewer unsaturated fatty acids are less susceptible to lipid peroxidation, which leads to less oxidation-mediated damage to macromolecules [39,96]. The metabolic significance of the MUFA/PUFA differences observed for the specific subclasses PI, LPE and Cer are not known to the best of the authors’ knowledge.

### 3.3. Moving Away from Subclass towards Species Analysis in Clinical Medicine

In clinical practice, lipid measurements are still often limited to HDL-C, LDL-C, total cholesterol and triglycerides. Although these parameters have been proven effective in evaluating cardiovascular risk, recent data demonstrated that specific ceramide species predict cardiovascular risk beyond them, calling for detailed lipid analysis at the molecular species level [10,26,29,92]. The results of the present study support this call considering that, in several cases, associations could be observed once zooming into species diversity within the same lipid subclass. For instance, no significant difference was observed for CE as a subclass between aged and young participants, while both CE(22:6) and CE(20:5) displayed significantly higher levels in aged compared to young participants. Different species within the same subclass have distinct biological roles, as illustrated by the cardiometabolic favourable and deleterious ceramide species [26,27,28,65]. Undoubtedly, lipidomic studies will provide new biomarkers to improve patients’ stratification and follow-up of patients suffering from cardiometabolic diseases [12].

### 3.4. Limitations

This study should be assessed in light of its limitations. First, investigating serum lipids does not provide information about their cellular origin, destination or subcellular localisation. Thus, the findings should be interpreted with caution, when it comes to mechanistic explanations. As this is an inherent limitation of cross-sectional population-based studies, the present results should be seen as a starting point for both prospective intervention and fundamental studies. Second, the cross-sectional nature of this study allows only for the establishment of associations, and not causality, between clinical and lipid phenotypes [97]. However, in light of the practical hurdles to conducting longitudinal studies over the lifespan, cross-sectional studies represent acceptable alternatives. Next, the biological significance of many lipid species is far from being understood, which prevents a comprehensive interpretation of the serum lipidome. Specifically, little or no data are available on the biological roles of GSL, LPE and LPI species [98,99,100,101,102]. In addition, the differences in lipid annotation levels also complicate data interpretation. For instance, many SM species were identified on the hydroxyl group level only, making a reflection in terms of Cer species difficult [103]. Another limitation is the fact that the biological insights provided by LION/web, the very first lipid ontology enrichment tool, are currently limited to associations between lipid species, physicochemical properties, general biological functions and organelles. Lastly, all participants lived in a small geographic area in Switzerland; therefore, results might not be generalisable to populations living in different regions of the world.

## 4. Materials and Methods

### 4.1. Participants

The investigated subset consisted of 73 young (25.2 ± 2.6 years, 43% female) and 77 aged (73.5 ± 2.3 years, 48% female) individuals. As reported in the study protocol, only healthy participants from the Basel area (Switzerland), who did not have exercise-limiting chronic disease, were non-smokers, or quit at least ten years ago, were included in the COmPLETE-Health study [54]. This excluded participants with a history of coronary artery disease, stroke, heart failure, lower-extremity artery disease, any kind of malignant tumour, diabetes, obesity, clinically apparent kidney failure, severe liver disease, chronic obstructive pulmonary disease GOLD stages two to four, arterial hypertension grades two and three, drug or alcohol abuse, exercise-limiting osteoporosis or orthopaedic conditions and clinically manifest Alzheimer’s disease or dementia. The study was conducted in accordance with the Declaration of Helsinki, approved by the Ethics Committee of North-Western and Central Switzerland (EKNZ 2017–01451) and registered on ClinicalTrials.gov (NCT03986892). All participants provided written informed consent.

### 4.2. Data Collection

Data were collected between January 2018 and June 2019. Prior to the clinical examination, participants were instructed not to diverge from habitual eating behaviour (for the previous 72 h), and to avoid exercising, drinking alcohol (for the previous 24 h) and drinking caffeinated beverages (for the previous 4 h). Participants were randomised in five time slots (08:00, 10:00, 12:00, 14:00 and 16:00) and measurements took approximately four hours each. After an hour of measurements at rest, trained medical staff collected blood samples in fasted state (at least three hours) by venepuncture of the cubital fossa (2 × 7.5 mL serum-gel, Monovette^®^, Sarstedt, Nümbrecht, Germany). Serum samples were gently shaken for 30 min, centrifuged (3000 rpm; 10 min; 20–23 °C), aliquoted and frozen at −80 °C.

Smoking status was assessed by telephone interview prior to the examination, while physicians reviewed medical history and medications by questionnaires on site. Body fat content was quantified using a four-segment bioelectrical impedance analysis (Inbody 720, Inbody Co. Ltd., Seoul, Korea). Physical activity was objectively monitored over the 14 consecutive days following the clinical examination using a wrist-worn triaxial accelerometer (GeneActive Activinsights Ltd., Kimbolton, UK). Data were analysed using the validated open-source Excel macro file “General physical activity” (version 2), quantifying total and moderate-to-vigorous physical activity in minutes per day (moderate defined as 4.00–6.99 Metabolic Equivalent of Task (METS) and vigorous ≥ 7 METS) [104]. The recruitment and data collection processes have been previously described in detail in the study protocol [54].

### 4.3. Biochemical Analysis

Total cholesterol, LDL-C, HDL-C and triglyceride concentrations were analysed from serum using an Olympus AU680 automatic analyser (Beckman Coulter, Brea, CA, USA), enzymatic reagents (DiaSys, Holzheim, Germany) and secondary standards (Roche Diagnostics, Mannheim, Germany). HbA1c was quantified from whole blood by high-pressure liquid chromatography using D-10 (Bio-Rad, Hercules, CA, USA).

### 4.4. Lipid Extraction

Lipids were extracted by the addition of 200 µL of Butanol/Methanol (1:1) solution to 40 µL of serum. Following the centrifugation for 15 min at 4000× *g* at 4 °C, the resulting supernatants were collected and transferred to liquid chromatography—mass spectrometry (LC-MS) vials for injection [105,106].

### 4.5. Untargeted Lipidomics

Serum lipid extracts were analysed by reversed-phase liquid chromatography coupled to a high-resolution mass spectrometry (RPLC-HRMS) instrument (Agilent 6550 iFunnel Q-TOF LC/MS, Agilent Technologies, Santa Clara, CA, USA) [107,108]. In both positive and negative ionisation mode, the chromatographic separation was carried out on a Zorbax Eclipse Plus C18 (1.8 μm, 100 mm × 2.1 mm I.D. column) (Agilent Technologies, Santa Clara, CA, USA). Mobile phase was composed of A = 60:40 (*v/v*) Acetonitrile:water with 10 mM ammonium acetate and 0.1% acetic acid and B = 88:10:2 Isopropanol:acetonitrile:water with 10 mM ammonium acetate and 0.1% acetic acid. The flow rate was 600 μL/min, column temperature 60 °C and sample injection volume 2 µL. Electrospray ionisation source conditions were set as follows: dry gas temperature 200 °C, nebuliser 35 psi and flow 14 L/min, sheath gas temperature 300 °C and flow 11 L/min, nozzle voltage 1000 V, and capillary voltage +/− 3500 V. Full scan acquisition mode in the range of 100–1700 mass-to-charge ratio (*m/z*) was applied for data acquisition while iterative tandem mass spectrometry (MS/MS) data-dependent acquisition at 25 eV was used to acquire the MS/MS data on pooled quality control (QC) samples. Iterative MS/MS was performed in five consecutive injections using computer-driven exclusion. The scan rate was set to three spectra/s with a duration of 333.3 ms/spectrum and a narrow isolation width of 1.3 *m/z*. The mass error tolerance was +/− 20 ppm with a retention time exclusion tolerance of +/− 0.1 min. Precursor ions were excluded after two spectra, with a maximum of three precursors per cycle. The precursor threshold was set to an absolute threshold of 5000 counts in positive mode and 2500 counts in negative mode.

### 4.6. Quality Control

Pooled QC samples (representative of the entire sample set) were analysed every ten samples throughout the overall analytical run in order to assess the quality of the data, correct the signal intensity drift and remove the peaks with poor reproducibility (CV > 20%) [109,110]. In addition, a series of diluted quality controls (dQC) were prepared by dilution with buthanol:methanol (1:1): 100%QC, 50%QC, 25%QC, 12.5%QC and 6.25%QC and analysed at the beginning and at the end of the sample batch. This dQC series served as a filter to remove the features, for which MS signal response was not linear (correlation with dilution factor < 0.8) [111].

### 4.7. Data Processing and Lipid Annotation

Raw RPLC-HRMS(/MS) data were deconvoluted using MS-DIAL version 4.00 and lipids were annotated by matching accurate mass, isotope ratio and MS/MS spectra with the LipidBlast library on 29 November 2019 [107,112]. Relative quantification of lipids was based on extracted ion chromatogram areas for the monitored precursor ions at the MS level. The obtained tables (containing peak areas of detected and annotated lipids by MS and MS/MS and MS only) were exported to an R-based in-house developed application, where the signal intensity drift over time was corrected using the locally estimated smoothing function (LOESS) and the cubic spline algorithm, followed by analytical variability filter (CV (QC features) > 20%) and visual inspection of linear response [109,110]. Redundancies were then removed. Finally, lipids identified on MS level only were targeted to obtain an MS/MS match.

### 4.8. Lipid Shorthand Notation

Using the LipidLynxX Converter tool, lipid shorthand notation issued from MS-DIAL was converted to the community-accepted shorthand notation system introduced by Liebisch et al. [103,113]. Two authors (J.C. and H.G.-A.) manually double-checked the shorthand notation of each lipid species.

### 4.9. Statistical Analysis

Each lipid species was assigned its corresponding lipid subclass, according to the LIPID MAPS^®^ Lipid Classification System, and saturation level of their aliphatic chains (saturated fatty acid, MUFA or PUFA) [114]. Lipid species abundances were summed to obtain abundances for each lipid subclass and saturation level. Abundances were log_2_-transformed prior to statistical analysis. MUFA to PUFA and PC to PE ratios were calculated on subclass levels, if applicable. Unless otherwise specified, all statistical analyses were conducted on the species and subclass levels as well as on MUFA to PUFA and PC to PE ratios.

To investigate global age-related differences in lipid species, unsupervised and supervised analyses were carried out using BIOMEX version 1.0–3, a novel web-based bioinformatic tool designed to facilitate the Biological Interpretation Of Multi-omics Experiments [115]. Specifically, dimensionality reduction was conducted using PCA, while differential analysis was computed using the linear limma model adjusted for blood sampling time [116].

Multiple linear regressions were run to assess associations between lipids, age, sex and the interaction between age and sex, while adjusting for the following previously described clinical confounders: body fat (%), statins intake, HbA1c (%), daily total physical activity (min), blood sampling time and fasting time prior to blood sampling [117,118,119,120,121,122,123,124,125]. Lipids were used as dependent variables, while the other parameters served as independent variables. Continuous dependent and independent variables were z-standardised prior to calculation [126]. Post-hoc tests, using the emmeans R-package (version 1.4.8), were calculated to determine and compare the estimated marginal means of each lipid in aged and young participants within both sexes [57]. The same was done for females and males in both age groups. Resulting β coefficients were converted to percentage of difference for ease of interpretation [57,127].
Percentage of difference = (2^β^ − 1) × 100(1)

Finally, lipid species were ordered by decreasing percentage of differences for all four age and sex groups. Resulting lists were entered into a lipid ontology enrichment tool (LION/web, version 2020.07.14), a web application developed to perform network analysis within complex lipidomic datasets in order to bridge the gap between data generated by lipidomic assays and their implication in cellular metabolism [58].

Normality of the residuals was checked graphically prior to running the above-mentioned statistical tests. For each statistical test, all *p*-values were adjusted together using the Benjamini-Hochberg method [128]. BIOMEX also uses the Benjamini–Hochberg method, while LION/web uses the q-value method to adjust *p*-values [58,115]. Adjusted *p*-values ≤ 0.05 were considered significant. Unless otherwise specified, statistical analyses were carried out using R (version 4.0.2) [129]. Rain plots were computed using a previously published R-code [130].

## 5. Conclusions

This study provides a comprehensive serum lipid profiling of clinically healthy humans in their twenties and seventies, allowing for the identification of age- and sex-associated lipid species. Age was identified as the major determinant of the circulating lipidome composition. Compared to the younger individuals, aged females and males exhibited significantly higher levels of sphingolipid, glycerophospholipid, cholesterol ester and triglyceride species. In particular, they showed higher levels in specific SM, Cer, PC, LPC, PI and CE species previously described as cardiometabolically favourable. These favourable features might indicate that aged participants benefited from protective genetic, environmental or lifestyle factors to reach their age in good health. Simultaneously, elevated levels of deleterious Cer, PC, PI and TGs species were also observed within the aged participants. This could sign a beginning age-related impairment in lipid metabolism. Remarkably, aged females were characterised by higher levels in several ether-glycerophospholipid and LPC species, which might support a potentially healthier age-related evolution of the lipidome composition within females. Finally, lipid species analysis revealed findings not captured by subclass analysis. This highlights the necessity to implement detailed lipid investigation at the molecular species level in clinical research studies in order to improve patients’ stratification and healthspan extension strategies.

## Figures and Tables

**Figure 1 metabolites-11-00287-f001:**
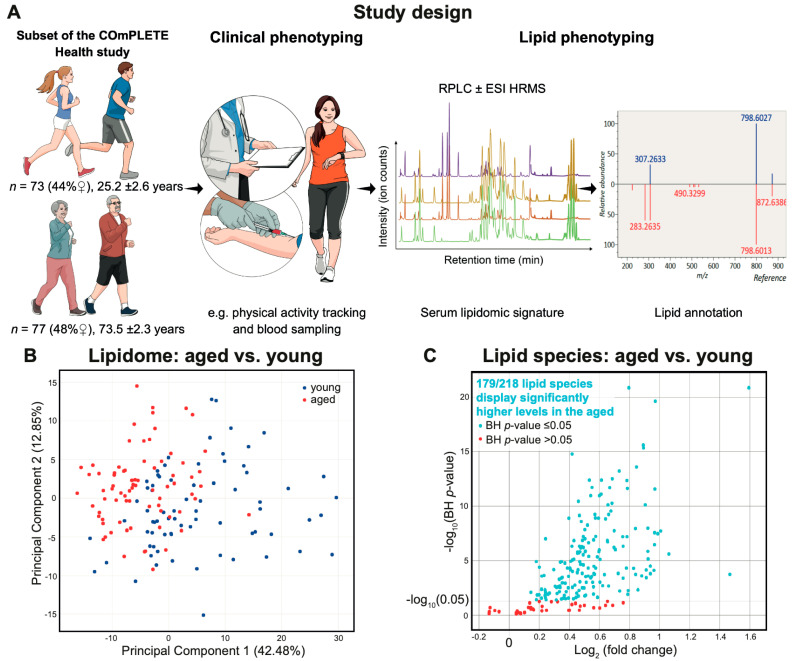
Study design, principal component analysis and differential analysis. (**A**) Overview of the study design including physical activity tracking and lipid phenotyping. (**B**) Principal component analysis highlighting the difference in lipid profile between young and aged participants. (**C**) Differential analysis highlighting lipid species displaying significantly higher levels (adjusted *p*-value ≤ 0.05) in the aged compared to the young participants after correction for blood sampling time. Abbreviations: RPLC ± ESI HRMS = Reversed-Phase Liquid Chromatography Electrospray Ionisation High-Resolution Mass Spectrometry; *m/z* = mass to charge ratio; BH = Benjamini–Hochberg. Panel A was created in the Mind the Graph platform (www.mindthegraph.com (accessed on 15 July 2020)).

**Figure 2 metabolites-11-00287-f002:**
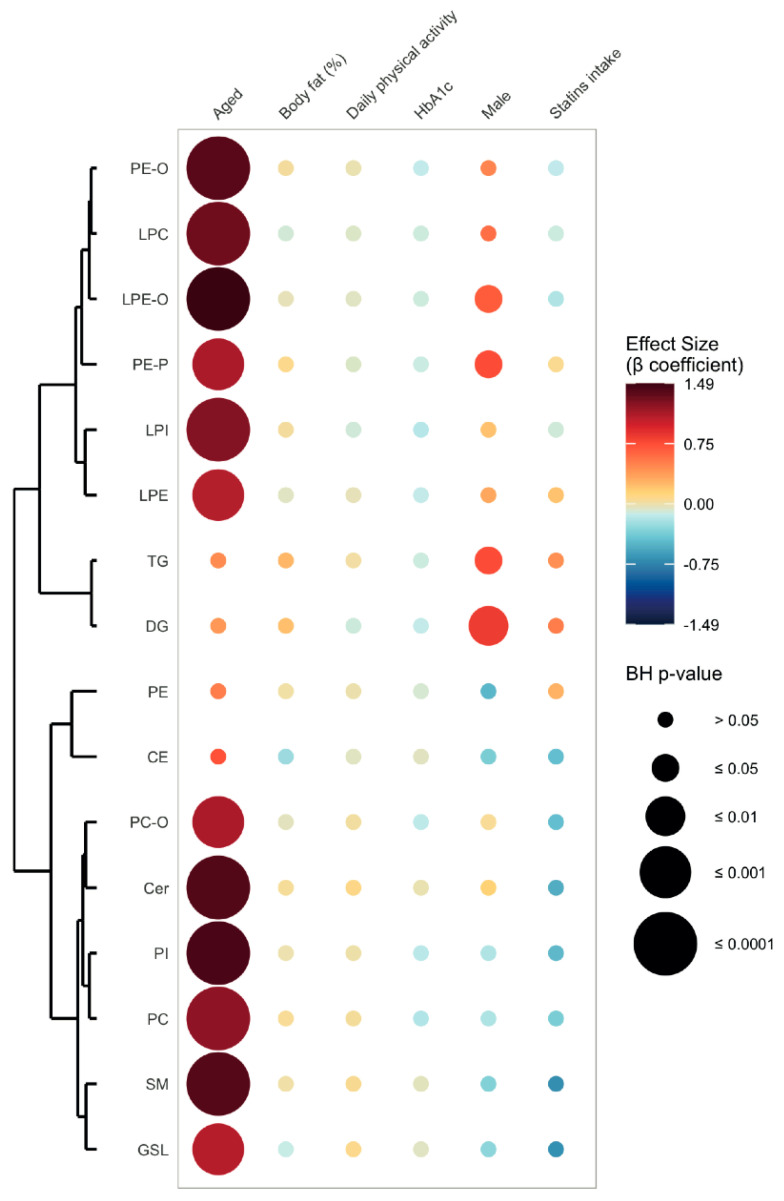
Associations between lipid subclasses, age, sex and clinical variables. Abbreviations: BH = Benjamini-Hochberg, DG = diglycerides, TG = triglycerides, CE = cholesterol esters, LPC = lyso-glycerophosphocholines, PC = glycerophosphocholines, PC-O = alkyl-glycerophosphocholines, LPE = lyso-glycerophosphoethanolamines, LPE-O = lyso-alkyl-glycerophosphoethanolamines, PE = glycerophospoethanolamines, PE-O = alkyl-glycerophosphoethanolamines, PE-P = alkenyl-glycerophosphoethanolamines, LPI = lyso-glycerophosphoinositols, PI = glycerophosphoinositols, Cer = ceramides, GSL = glycosphingolipids, SM = sphingomyelins, MUFA = monounsaturated fatty acid, PUFA = polyunsaturated fatty acids, HbA1c = glycated haemoglobin.

**Figure 3 metabolites-11-00287-f003:**
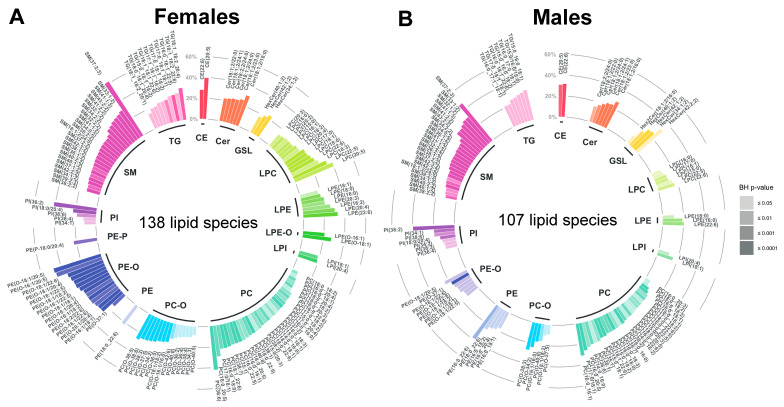
Lipid species: percentage of difference in the aged compared to the young. (**A**) Females. (**B**) Males. Abbreviations: BH = Benjamini-Hochberg, CE = cholesterol ester, Cer = ceramide, GSL = glycosphingolipids, LPC = lyso-glycerophosphocholines, LPE = lyso-glycerophosphoethanolamines, LPE-O = lyso-alkyl-glycerophosphoethanolamines, LPI = lyso-glycerophosphoinositols, PI = glycerophosphoinositols, PC = glycerophosphocholines, PC-O = alkyl-glycerophosphocholines, PE = glycerophospoethanolamines, PE-O = alkyl-glycerophosphoethanolamines, PI = glycerophosphoinositols, SM = sphingomyelins and TG = triglycerides.

**Figure 4 metabolites-11-00287-f004:**
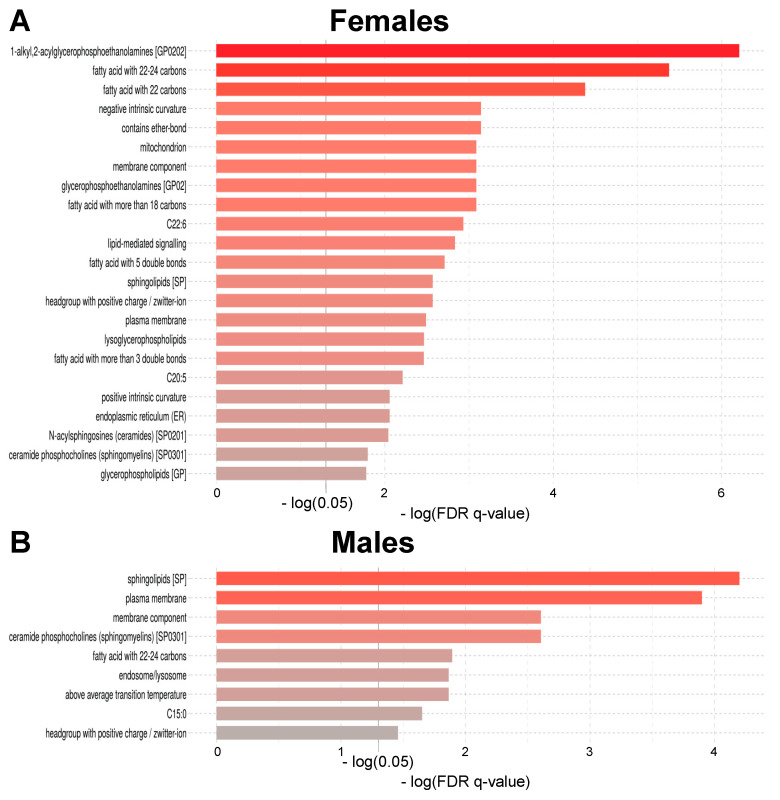
Lipid ontology enrichment analysis: aged compared to young. (**A**) Females. (**B**) Males. Abbreviations: FDR = false discovery rate.

**Figure 5 metabolites-11-00287-f005:**
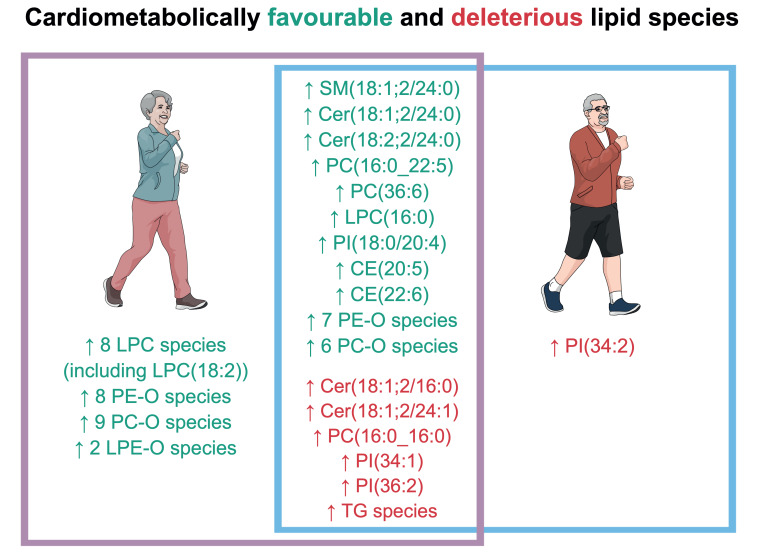
Summary of the main findings. Abbreviations: SM = sphingomyelins, Cer = ceramide, PC = glycerophosphocholines, LPC = lyso-glycerophosphocholines, PI = glycerophosphoinositols, CE = cholesterol ester, PE-O = alkyl-glycerophospoethanolamines, PC-O = alkyl-glycerophosphocholines, LPE-O = lyso-alkyl-glycerophosphoethanolamines, TG = triglycerides. Figure 5 was created in the Mind the Graph platform (www.mindthegraph.com (accessed on 15 February 2021)).

**Table 1 metabolites-11-00287-t001:** Participants’ characteristics.

	Young		Aged	
	Female	Male	Female	Male
**Participants, *n* (%)**	32 (21.3)	41 (27.3)	37 (24.7)	40 (26.7)
**Anthropometry, mean ± SD**				
Age (years)	25.1 ± 2.3	25.1 ± 2.8	74.0 ± 2.4	73.9 ± 2.5
Body mass (kg)	60.5 ± 9.0	76.7 ± 9.5	61.8 ± 7.5	74.9 ± 8.3
Body fat mass (%)	23.1 ± 6.9	14.7 ± 5.3	30.4 ± 7.5	24.3 ± 6.1
Body mass index (kg/m^2^)	21.5 ± 2.9	23.7 ± 2.3	23.5 ± 3.0	24.9 ± 2.6
Systolic blood pressure (mmHg)	111 ± 8	126 ± 10	137 ± 12	133 ± 13
Diastolic blood pressure (mmHg)	71 ± 8	71 ± 7	80 ± 8	82 ± 8
**Smoking status, *n* (%)**				
Never smoked	31 (97)	40 (98)	26 (70)	21 (52)
Ex-smokers (quit > 10 years ago)	1 (3)	1 (2)	11 (30)	19 (48)
**Physical activity (PA) levels, mean ± SD**				
Daily total PA (min)	282.3 ± 56.1	274.7 ± 69.3	257.2 ± 87.3	237.7 ± 75.2
Daily moderate-to-vigorous PA (min)	190.7 ± 45.1	186.7 ± 53.1	142.4 ± 63.0	140.8 ± 55.3
**Biochemical parameters, mean ± SD**				
Fasting time prior to blood sampling (h)	6.0 ± 1.6	5.6 ± 2.0	6.6 ± 3.7	7.4 ± 4.5
Total cholesterol (mmol/L)	4.96 ± 0.78	4.68 ± 0.95	6.49 ± 0.80	5.96 ± 1.11
LDL-C (mmol/L)	2.62 ± 0.48	2.52 ± 0.56	3.56 ± 0.59	3.40 ± 0.73
HDL-C (mmol/L)	1.81 ± 0.42	1.43 ± 0.24	1.91 ± 0.35	1.58 ± 0.33
Triglycerides (mmol/L)	1.08 ± 0.53	1.33 ± 0.80	1.41 ± 1.06	1.35 ± 0.45
HbA1c (%)	5.0 ± 0.2	5.0 ± 0.2	5.4 ± 0.3	5.3 ± 0.3
**Cardiovascular medications, *n* (%)**				
Antihypertensives	0 (0)	0(0)	9 (16)	19 (35)
Low-dose aspirin	0 (0)	0 (0)	2 (5)	4 (10)
Statins	0 (0)	0 (0)	3 (8)	6 (15)
**Hormonal medications, *n* (%)**				
Oestrogen/HRT	4 (13)	0 (0)	5 (14)	0 (0)
5α-reductase inhibitors	0 (0)	0 (0)	0 (0)	4 (10)
Thyroid hormones	0 (0)	0 (0)	3 (8)	3 (8)
**Psychiatric medications, *n* (%)**				
Antidepressants	1 (3)	1 (2)	1 (3)	0 (0)
Z-drugs	0 (0)	0 (0)	3 (8)	0 (0)
**Other medications, *n* (%)**	3 (9)	6 (12)	31 (49)	24 (42)

Abbreviations: LDL-C = low-density lipoprotein cholesterol, HDL-C = low-density lipoprotein cholesterol, HbA1c = glycated hemoglobin, HRT = hormone replacement therapy, Z-drugs = nonbenzodiazepine benzodiazepine receptor agonists. Other drugs include: vitamins (19), nonsteroidal anti-inflammatory drugs (8), proton-pump inhibitors (7), topical ophthalmic drugs (7), antihistamines (4), tamsulosin (3), melatonin (2), anthocyanosides of vaccinium myrtillus (1), clindamycin (1), fluticasone/salmeterol (1), fluticasone/vilanterol (1), ibandronate (1), mesalazine (1), mometasone (1), paracetamol (1), prednisolone (1), pregabalin (1), tiotropium (1), zoledronate (1).

**Table 2 metabolites-11-00287-t002:** Identified lipid species and their respective lipid subclass.

Lipid Subclass, Full Name	Lipid Subclass, Abbreviation	Identified Lipid Species, *n*
Diglycerides	DG	2
Triglycerides	TG	58
Cholesterol esters	CE	5
Glycerophosphocholines	PC	42
Alkyl-glycerophosphocholines	PC-O	16
Lyso-glycerophosphocholines	LPC	15
Glycerophosphoinositols	PI	7
Lyso-glycerophosphoinositols	LPI	2
Glycerophospoethanolamines	PE	11
Alkyl-glycerophosphoethanolamines	PE-O	15
Alkenyl-glycerophosphoethanolamines	PE-P	1
Lyso-glycerophosphoethanolamines	LPE	7
Lyso-alkyl-glycerophosphoethanolamines	LPE-O	2
Ceramides	Cer	6
Sphingomyelins	SM	24
Glycosphingolipids	GSL	5

## Data Availability

All data presented in this study are available within the article and Appendix A.

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
