# Peer review of "Metabolic View on Human Healthspan: A Lipidome-Wide Association Study"

_metabolites, 2021, doi:10.3390/metabo11050287_

Round 1
Reviewer 1 Report
The manuscript by Carrard et al compares serum lipid profiles between young and old healthy persons. They use a mass spec approach and detect 218 lipid species that they then compare between young and old. The study purposefully profiles lipid changes in healthy subjects to assess normal aging in the absence of clinical pathology. Of note, there appear to be only positive correlations between lipid abundance and aging, lipid concentration either remains steady or goes up with age. Lipid changes in age are very pronounced, while body fat, activity, sex, or even statin intake have much smaller effects. Finally, the authors discuss the changes in lipid abundance in the light of the literature, grouping lipids in favorable and unfavorable groups. Summarizing, the authors suggest that knowing normal age-related changes in serum lipids provides a baseline for future clinical evaluation of patients based on many lipid species instead of just the current cholesterol species.
This is a descriptive study that has potential relevance as a reference for future clinical lipid profiling in people. However, it is lacking biological insights at the current stage.
Major points
- It is not clear to me what the result and interpretation of the LION analysis are. Why was this analysis done and what do the authors learn from it? What is the relevance of the high-scoring processes?
- While it is a great approach to follow healthy aging, it would be very informative to compare the healthy age-related changes with the lipid levels in pathological conditions. This could help identify which lipid changes that occur in normal healthy aging might also be predictive for disease. This would have aspects of a biomarker and add a lot of significance to the paper.
Minor points
- In the abstract, it is not clear what lipid species and subclasses are referred to. It might help to add more information for the naïve reader.
- In the abstract, it is not clear why ether-glycerophospho-lipid and lyso-glycerophosphocholine species might be favorable. It would help the understanding to explain in what context this was found.
- Line 53: hundreds of thousands of lipid species. Is it really more than 100.000?
- Line 127: What was the method to determine which lipids to measure? This would be important to note in the results. Was this an unbiased approach, or were the lipids for quantification selected before the study? In other words, are these the most common or abundant lipids? This would be important as there are at least 600 lipids and would be important to know how representative the detected lipids are.
Reviewer 2 Report
The authors present a comprehensive and solid study about lipid profiling on healthy individuals and the identification of age- and sex-associated lipid species.
The study is well designed. The authors present sufficient data to support the association of lipid species with age and sex.
I recommend accepting the manuscript in the present form.
Reviewer 3 Report
The important role of lipids in cellular metabolism is very well known; the identification of hundreds to thousands of lipids in plasma and tissue extracts by mass spectrometry-based lipidomics techniques has helped in better understanding their implications in health and disease.
The current paper is very interesting; the authors have meticulously studied lipidome in older people and compared it to a younger group and they have performed multiple linear regressions (adjusting also for known confounders) in an effort to identify how lipidome naturally varies by age and sex in healthy individuals, a still not well known factor.
The paper is well organized, the study is well designed and the data are clearly presented adding information in the biology of elderly. The limitations highlight the necessity for further studies in order to elucidate important processes involved in healthy ageing and longevity. relative disease mechanisms or disease-related biomarkers.
Round 2
Reviewer 1 Report
I appreciate the author's comments and the changes made in the manuscript. I support publication of the paper in the present form.